# *GBA1*-Associated Parkinson’s Disease Is a Distinct Entity

**DOI:** 10.3390/ijms25137102

**Published:** 2024-06-28

**Authors:** Aliaksandr Skrahin, Mia Horowitz, Majdolen Istaiti, Volha Skrahina, Jan Lukas, Gilad Yahalom, Mikhal E. Cohen, Shoshana Revel-Vilk, Ozlem Goker-Alpan, Michal Becker-Cohen, Sharon Hassin-Baer, Per Svenningsson, Arndt Rolfs, Ari Zimran

**Affiliations:** 1Rare Disease Consulting RCV GmbH, Leibnizstrasse 58, 10629 Berlin, Germany; 2Shmunis School of Biomedicine and Cancer Research, Faculty of Life Sciences, Tel Aviv University, 6997801 Ramat Aviv, Israel; 3Gaucher Unit, Shaare Zedek Medical Center, 9103102 Jerusalem, Israel; 4Agyany Pharma Ltd., 9695614 Jerusalem, Israel; 5Translational Neurodegeneration Section Albrecht Kossel, Department of Neurology, University Medical Center Rostock, 18147 Rostock, Germany; 6Center for Transdisciplinary Neurosciences Rostock (CTNR), University Medical Center Rostock, University of Rostock, 18147 Rostock, Germany; 7Department of Neurology and Movement Disorders Unit, Shaare Zedek Medical Center, 9103102 Jerusalem, Israel; 8Faculty of Medicine, Hebrew University of Jerusalem, 9112102 Jerusalem, Israel; 9Lysosomal and Rare Disorders Research and Treatment Center, Fairfax, VA 22030, USA; 10Movement Disorders Institute, Department of Neurology, Chaim Sheba Medical Center, 5262101 Tel-Hashomer, Israel; 11Department of Neurology and Neurosurgery, Faculty of Medical and Health Sciences, Tel Aviv University, 6997801 Tel-Aviv, Israel; 12Department of Clinical Neuroscience, Karolinska Institute, 17177 Stockholm, Sweden; 13Department of Basal and Clinical Neuroscience, King’s College London, London SE5 9RT, UK; 14Medical Faculty, University of Rostock, 18055 Rostock, Germany

**Keywords:** Parkinson’s disease, *GBA1* variants, *GBA1*-associated Parkinson disease, clinical presentation and course, genotype-phenotype correlations, pathophysiology and molecular mechanisms, treatment options

## Abstract

*GBA1*-associated Parkinson’s disease (*GBA1*-PD) is increasingly recognized as a distinct entity within the spectrum of parkinsonian disorders. This review explores the unique pathophysiological features, clinical progression, and genetic underpinnings that differentiate *GBA1*-PD from idiopathic Parkinson’s disease (iPD). *GBA1*-PD typically presents with earlier onset and more rapid progression, with a poor response to standard PD medications. It is marked by pronounced cognitive impairment and a higher burden of non-motor symptoms compared to iPD. Additionally, patients with *GBA1*-PD often exhibit a broader distribution of Lewy bodies within the brain, accentuating neurodegenerative processes. The pathogenesis of *GBA1*-PD is closely associated with mutations in the *GBA1* gene, which encodes the lysosomal enzyme beta-glucocerebrosidase (GCase). In this review, we discuss two mechanisms by which *GBA1* mutations contribute to disease development: ‘haploinsufficiency,’ where a single functional gene copy fails to produce a sufficient amount of GCase, and ‘gain of function,’ where the mutated GCase acquires harmful properties that directly impact cellular mechanisms for alpha-synuclein degradation, leading to alpha-synuclein aggregation and neuronal cell damage. Continued research is advancing our understanding of how these mechanisms contribute to the development and progression of *GBA1*-PD, with the ‘gain of function’ mechanism appearing to be the most plausible. This review also explores the implications of *GBA1* mutations for therapeutic strategies, highlighting the need for early diagnosis and targeted interventions. Currently, small molecular chaperones have shown the most promising clinical results compared to other agents. This synthesis of clinical, pathological, and molecular aspects underscores the assertion that *GBA1*-PD is a distinct clinical and pathobiological PD phenotype, necessitating specific management and research approaches to better understand and treat this debilitating condition.

## 1. Introduction

Parkinson’s disease (PD) is a complex neurodegenerative disorder characterized by the progressive loss of dopaminergic neurons, predominantly in the Substantia Nigra pars compacta (SNpc), accompanied by the presence of Lewy bodies (LBs), which are fibrillar aggregates primarily composed of alpha-synuclein. While the majority of PD cases are idiopathic (iPD) with no identifiable genetic mutation or known cause, a subset of patients present with monogenic forms of PD, among them is *GBA1*-associated PD (*GBA1*-PD). Mutations in the *GBA1* gene, which encodes the lysosomal enzyme glucocerebrosidase (GCase), constitute the most significant genetic risk factor for developing PD.

In this review, we examine the distinctive clinical, pathological, and molecular characteristics of *GBA1*-PD. We also consider the implications of *GBA1* variants in therapeutic strategies, emphasizing the importance of early diagnosis and targeted interventions. Through this synthesis, we aim to provide a comprehensive understanding of *GBA1*-PD, affirming its status as a unique clinical and pathobiological PD phenotype.

## 2. PD Clinical Manifestations and Disease Course

### 2.1. PD Definition

PD is clinically defined by the presence of cardinal motor symptoms, bradykinesia in combination with at least one rest tremor or rigidity [1]. The cardinal motor symptoms depend on the progressive degeneration of the dopamine-containing neurons in the SNpc [2]. The histopathological hallmark of PD is the presence of Lewy bodies (LBs), fibrillar aggregates in which alpha-synuclein is the major constituent [3]. Pathological studies have shown a strong correlation between the extent of LB-related cell loss in the SNpc and the severity of bradykinesia [4]. Pathological studies estimate a 40–60% loss of dopaminergic cells and a significant reduction in synaptic function before the appearance of motor symptoms meets the current PD diagnostic criteria [4,5]. The spread of LBs by the caudal to rostral pattern is associated with early involvement of the peripheral autonomic nervous system [6].

### 2.2. Clinical Manifestations of GBA1-PD

The pathological characteristics of *GBA1*-PD might appear as mirror images of those seen in iPD, including nigrostriatal dopamine depletion and accumulation of alpha-synuclein aggregates, forming LBs within the brainstem and cortex [7,8,9,10,11,12,13]. However, it has been shown that patients with *GBA1*-PD display a broader distribution of LBs across the brain compared to iPD [14]. In the histology of *GBA1*-PD, there is a co-localization of the mutant GCase along with alpha-synuclein, presumably in lysosomes, and it has been speculated that this accumulation contributes to the further aggregation of the alpha-synuclein, thereby leading to the more severe clinical consequences [15].

Although the clinical symptoms of *GBA1*-PD can resemble those of iPD, some studies have reported that subsets of *GBA1*-PD can be particularly benign, with a mild course and relatively preserved cognition [16,17], and without significant differences in olfaction between the *GBA1*-PD and iPD groups [18]. Most studies have reported a distinct clinical profile of *GBA1*-PD compared to iPD, including earlier onset, faster progression, and more severe motor and non-motor symptoms. *GBA1*-PD is associated with greater cognitive decline, a higher prevalence of mood disorders, and more severe sleep disturbances. Non-motor symptoms, such as olfactory dysfunction and autonomic issues, are also more pronounced in patients with *GBA1*-PD (Table 1).

### 2.3. GBA1-PD Genotype-Phenotype Correlations

The intricate relationship between *GBA1* variants and their clinical manifestations in PD and Gaucher Disease (GD) underscores the complex interplay between genetic mutations and neurodegenerative pathologies. Different variants of the *GBA1* gene result in varying levels of residual activity of the mutant GCase enzyme compared to the enzyme produced by the wild-type (WT) *GBA1* gene. In certain variants, the enzymatic activity of GCase is considerably high; for instance, the p.D140H variant retains 60–73% of the activity found in WT GCase. Conversely, other variants exhibit only residual activity, such as the p.P415R variant, which shows only 0–4.3% of WT GCase activity [44].

A comprehensive analysis of *GBA1* variants in the ClinVar database (https://www.ncbi.nlm.nih.gov/clinvar/ accessed on 24 June 2024) reveals a total of 383 identified variants. The pathogenicity of these variants is diverse, with 111 classified as pathogenic, 92 as likely pathogenic, 131 as of uncertain significance, 37 as likely benign, and 11 as benign. Additionally, 36 variants have conflicting classifications. The majority of these variants are missense mutations (242), which involve a single nucleotide change that results in a different amino acid in the protein. Other types of mutations include 23 frameshift variants, 23 nonsense variants, and 8 splice site variants. There are also 30 UTR variants reported. In terms of the variation type, single-nucleotide variants dominate the list, with 331 occurrences. There are also 33 deletions, 18 duplications, 7 insertions, and 2 indels. Most of the variants are short, with less than 50 base pairs (359), and only 8 are classified as structural variants (≥50 base pairs). In another comprehensive data source, the *GBA1* Variant Browser (https://pdgenetics.shinyapps.io/GBA1Browser/ accessed on 24 June 2024), 371 *GBA1* variants are categorized based on their association with GD severity and risk for PD [45]. Detailed information regarding the *GBA1* variants most commonly associated with PD is presented in Table 2.

The catalytic enzymatic domain of the *GBA1* gene, which encodes GCase, is specifically located within exons 2 to 10. This region is crucial for the enzyme’s function, as it includes the active site residues necessary for its catalytic activity [46,47]. The proper functioning of this domain is essential for the breakdown of glucocerebrosides in lysosomes, and mutations within these exons are often associated with GD and PD.

Specifically, 23 variants are classified as mild, being linked to type 1 GD (GD1), while 84 variants are considered severe, associated with type 2 (GD2) and type 3 (GD3) GD [45].

The variants, such as p.T369M and p.E388K, were originally defined as not causing GD but are recognized as risk factors for PD. Despite the p.E326K variant traditionally being considered non-causative for GD, recent studies have revealed a more nuanced role [44,48,49]: contrary to prior assumptions, p.E326K does significantly impact the enzymatic activity of GCase, retaining approximately 21–25% of WT GCase activity; cells expressing the p.E326K variant demonstrate a significant increase in lipid droplet formation, evidencing a 2.11-fold increase compared to WT *GBA1*. Recent data confirm that the p.E326K variant can cause disease manifestations in homozygous settings, as well as in compound heterozygous scenarios, particularly when paired with severe mutations like p.L444P [50]. These findings challenge the earlier classification of p.E326K as noncontributory to GD and underscore its potential significance in specific genetic contexts. The majority of variants, 262, could not be classified due to insufficient information on their relationship to GD pathology or their risk of leading to PD [45].

Generally, the severity of *GBA1* mutations in GD correlates with the severity of PD and inversely correlates with GCase activity [51]. Severe mutations are associated with an earlier PD onset and increased disease risk [8,19,52], characterized by a higher symptom burden, cognitive decline, and dementia risk.

Severe *GBA1* variants, such as p.L444P and p.R120W, associated with neuronopathic GD (nGD), significantly increase PD risk and hasten disease progression [53]. Patients with a severe *GBA1* variant exhibit a heightened PD risk and an earlier onset than those with mild variants [8,22,54]. Dopamine transporter imaging reveals that severe variants lead to a more significant loss of presynaptic dopaminergic terminals, indicating a dose-dependent relationship between *GBA1* genotypes and PD severity [54].

The risk of developing PD decreases in a gradient from severe to mild GD variants and further to non-GD-causing variants of the *GBA1* gene [45]. The odds ratios (OR) for PD development range from 2.84 to 4.94 for mild and 9.92 to 21.29 for severe GD variants. However, this trend is not always consistent. For instance, some mild GD variants, such as the p.N370S, are associated with a higher risk of developing PD (OR for PD risk: 11.40) [52] compared to certain severe variants, like p.V394L (OR ranges from 4.85 to 6.70) [52,55]. However, this inconsistency might be related to the higher frequency of the p.N370S variant compared to other GD variants. Moreover, variants such as p.T369M and p.E388K do not manifest GD clinically in a homozygous state but increase PD risk in both homozygous and heterozygous forms in some, but not all, populations [56,57,58,59]. The p.E326K, mild GD variant, one of the most prevalent *GBA1* variants among patients with PD [44,60,61,62], is associated with a severe PD phenotype [49,63,64,65]. The variants, p.T369M, and p.E388K have been potentially linked to an earlier PD disease onset and a faster progression of both motor and cognitive decline [7,33,63,64,66,67]. A recent meta-analysis highlighted the p.E326K variant’s role in significantly increasing cognitive impairment risk among patients with PD, while p.T369M is linked to accelerated motor disability progression [68].

**Table 2 ijms-25-07102-t002:** Most common variants in the *GBA1* gene associated with PD (Transcript ID: NM_000157.4).

Variant *	Exon	c.DNA	Impact	GD ACMG	GD Severity	OR PD	gnomAD #	Ref.
p.N370S	9	c.1226A > G	missense	P	mild	1.8–7.8	0.0022349	[45,52,56,57,58,59]
p.L444P	10	c.1448T > C	missense	P	severe	2.6–30.4	0.0012957	[45,50,53]
p.E326K	8	c.1093G > A	missense	B	risk PD variant	1.6–3.3	0.0107311	[44,45,48,49,50,60,62]
p.T369M	9	c.1223C > T	missense	VUS	risk PD variant	1.4–5.0	0.0061242	[44,56,57,58,59,68]
p.D409H	9	c.1342G > C	missense	P	severe	-	0.0001278	[44,45]
p.R496H	11	c.1604G > A	missense	P	mild	3.2–4.4	0.0001694	[44,45]
p.R120W	4	c.475C > T	missense	P	severe	-	<0.0000001	[44,45,53]

PD—Parkinson’s disease; GD—Gaucher Disease; ACMG—American College of Medical Genetics and Genomics Classification; P—pathogenic; B—benign; VUS—a variant of uncertain significance; OR PD—odds ratios (OR) for PD development; *—the old nomenclature is used (e.g., p.N370S) instead of the new one (e.g., p.Asn409Ser) since most of the cited sources also use old nomenclature; #—overall allele frequency (gnomAD v2.1.1); ‘-’ indicates no data available.

## 3. Epidemiology

### 3.1. PD Epidemiology

PD impacts 1–2% of individuals over 65 years of age, with a prevalence rising to 5% among those over 85 [69]. As the global population ages, the incidence of PD is rapidly increasing, making it one of the fastest-growing neurological disorders in terms of disability and mortality. The prevalence of PD has notably doubled over the past 25 years [70], and these figures likely underestimate the true impact, as they do not account for the various forms of parkinsonism, including atypical parkinsonism.

The World Health Organization (WHO) reported that, in 2017, the global annual incidence rate of PD was 1.02 million. By 2019, over 8.5 million individuals worldwide were living with PD. This resulted in 5.8 million disability-adjusted life years—an 81% increase since 2000—and led to 329,000 deaths, more than doubling since 2000 [71]. with the age-standardized rate (ASR) of prevalence increasing by 21.7% from 1990 to 2016 [70,72]. The International Parkinson and Movement Disorder Society estimated that, by 2020, around 9.4 million individuals were affected by PD globally (https://www.movementdisorders.org/ accessed on 24 June 2024).

In the United States (USA), nearly one million people are living with PD, with projections suggesting this number will increase to 1.2 million by 2030. The USA sees nearly 90,000 new PD diagnoses each year. The economic burden of PD in the USA., including healthcare costs, social security payments, and lost income, is estimated at nearly $52 billion annually [73,74].

In European countries (France, Germany, Italy, the Netherlands, Portugal, Spain, Sweden, and the United Kingdom), crude prevalence rate estimates range from 65.6 to 12,500 per 100,000, and annual incidence estimates range from 5 to 346 per 100,000 [75]. The most recent study conducted in Norway between 2005 and 2016 showed that the average crude incidence of PD was 23.1 for females and 29.6 for males per 100,000 person-years. The prevalence of PD in the population was, on average, 0.2% (200/100,000) for females and 0.23% (230/100,000) for males in the general population, and 0.98% for females, and 1.35% for males in the population of >65 years. For both sexes, the age-specific incidence and prevalence increased with age, peaking at the 75–85 age group [76].

However, data on the incidence and prevalence of PD in low- and middle-income countries (LMICs) and among ethnic minorities in high-income countries (HICs) are inconsistent [77,78,79,80,81]. This is due to financial and geographical healthcare access barriers, underreporting, misdiagnosis, and lack of awareness about PD. These challenges complicate efforts to accurately estimate PD’s global impact, and rigorous economic data on PD costs are scarce in most countries.

### 3.2. Epidemiology of GBA1-PD

Most PD cases likely have a multifactorial etiology involving a combination of environmental and genetic factors that influence alpha-synuclein aggregation and clearance [82,83]. A family history of PD increases the risk 3–4 times, suggesting a significant risk of genetic influence [84]. In recent decades, numerous monogenic forms of PD have been identified, with several genes associated with these forms cataloged in the Online Mendelian Inheritance in Man (OMIM) database (https://www.omim.org/entry/168600 accessed on 24 June 2024). Key genes involved include *SNCA*, *LRRK2*, and *GBA1*, among others.

*GBA1* variants are found in 2–31% of patients with PD. This makes *GBA1* mutations the most important genetic risk factor for PD identified to date. However, this number varies significantly across different ethnic groups, ranging between 2% and 12% in populations of non-Ashkenazi Jewish (n-AJ) origin to 10–31% in Ashkenazi Jewish (AJ), in contrast to less than 1% in the general population (3% among general AJ population). [46,53,85,86,87,88,89,90]. The proportion of those carrying *GBA1* variants is 2%–10.7% among Chinese patients with PD [91,92,93], 3.2% among South Korean patients [94], 9.4% among Japanese patients [95], and 2.9–8.0% among North and South American patients [8,88].

Zhao et al. analyzed 23 known PD genes in 1676 Chinese patients. The overall molecular diagnostic yield was 7.88%. The study revealed that *PRKN* emerged as the gene most commonly subject to mutations, with such alterations found in 4.95% of patients with PD. Variants in the *GBA1* gene were identified as the second most prevalent, detected in 3.64% of individuals in the cohort [96].

A genome-wide association study (GWAS) study recently identified a novel genetic risk factor, non-coding *GBA1* rs3115534 variant, in people of African ancestry, which has not been observed in European populations [97].

The Rostock Parkinson’s Disease Study (ROPAD) is the worldwide largest and most comprehensive PD prospective genetic screening study to date. It successfully overcomes the drawbacks of previous studies by investigating the presence of variants in 50 genes with established (*n* = 26) or possible (*n* = 24) relevance for parkinsonism among 12,580 patients with PD from 16 different countries, irrespective of their family history or putative inheritance patterns. The preliminary findings of the ROPAD study from 1288 patients, comprising 10% of the entire ROPAD patient group, resulted in 169 study participants (13.1%) with mutant variants in *GBA1* (8.5%), *LRRK2* (3.1%), and *PRKN* (0.8%) genes [87].

A comprehensive analysis of the complete ROPAD dataset (accepted for publication in Brain, 2024) revealed that 15% of all PD cases have at least one mutation in a PD-associated gene. Among those with positive genetic findings, the distribution of genetic variants included *GBA1* risk variants in 10.4% of the cases, pathogenic or likely pathogenic variants in *LRRK2* (2.9%), *PRKN* (0.9%), *SNCA* (0.2%), *GCH1* (0.2%), *PINK1* (0.1%), and a collection of 10 other genes (0.1%). Notably, approximately 0.6% of participants in the ROPAD study exhibited positive genetic testing results for genes associated with dystonia/dyskinesia (*TOR1A*, *SGCE*, *GNAL*, *THAP1*, and *KMT2B*) or dementia (*APP*, *PSEN1*, *GRN*, and *MAPT*), highlighting a genetic link to these conditions in the PD population.

Pathogenic variants in the *GBA1* gene have an age-dependent penetrance in PD, which is highly variable across different reports, ranging between 8% and 30% by the age of 80 years [26,98,99,100]. In a recent study, Balestrino et al. employed a kin-cohort approach to examine the penetrance of pathogenic *GBA1* variants among a cohort of patients with PD who were not preselected based on genetic risk. This study design, which considered both probands and their family members, allowed for a nuanced assessment of disease risk associated with these genetic markers. The findings reveal that individuals carrying these variants face an incremental risk of developing the disease by the ages of 60, 70, and 80 years, with respective probabilities of 10%, 16%, and 19% [101]. The studies also found a trend towards greater PD penetrance for severe pathogenic variants compared to mild pathogenic variants in the *GBA1* gene [101,102].

In summary, *GBA1* mutations substantially increase the risk of developing PD by 5–30 times, depending on the mutation severity and its prevalence in different ethnic groups [7,19,88]. Ongoing efforts aim to identify early-PD symptoms in large cohorts of subject carriers of variants in the *GBA1* gene to enable earlier diagnosis and prediction of PD onset [103]. Additionally, *GBA1* mutations have been linked to dementia with LBs, further underscoring the connection between *GBA1* mutations and alpha-synucleinopathies [10,104].

## 4. *GBA1*-PD Pathophysiology and Molecular Mechanisms

### 4.1. Alpha-Synuclein

The accumulation and aggregation of alpha-synuclein in the brain are key features of PD. Alpha-synuclein is a small, natively unfolded protein primarily found in the brain, although it is also present in smaller amounts in other tissues [105,106]. It plays a role in the normal functioning of neurons, particularly in synaptic vesicle turnover and neurotransmitter release [107,108,109,110,111]. Alpha-synuclein is synthesized in the cytoplasm of neurons and is then transported to its sites of action, including the presynaptic terminals. In the presynaptic terminals, alpha-synuclein is believed to play several roles in the regulation of neurotransmitter release. It is involved in the maintenance of a pool of synaptic vesicles, assisting in their trafficking and docking at the presynaptic membrane and possibly in the modulation of neurotransmitter release. Although its precise physiological functions are not fully understood, alpha-synuclein is thought to contribute to the efficient packaging and transport of neurotransmitters within vesicles as well as synaptic plasticity [112]. Alpha-synuclein can be degraded by the ubiquitin–proteasome system (UPS), a pathway that targets proteins for degradation by tagging them with ubiquitin molecules. This system is crucial for regulating protein turnover and eliminating misfolded or damaged proteins, including alpha [113,114]. Another pathway for alpha-synuclein degradation is the autophagy-lysosome pathway (ALP). During this process, alpha-synuclein is sequestered into autophagosomes, which then fuse with lysosomes, where the protein is degraded by lysosomal enzymes. This pathway is particularly important for the degradation of aggregated or oligomeric forms of alpha-synuclein [115]. Dysregulation of any part of this process can lead to either a decrease or an increase in alpha-synuclein levels [114,116]. Variants in genes that encode components of the UPS (e.g., *PARK2*, *UCHL1*, *FBXO7*) and ALP (e.g., *LAMP2A*, *PARK9*) or directly affect alpha-synuclein (i.e., *SNCA*) can impair its degradation, leading to accumulation and aggregation of the protein, which is toxic to neurons and further leads to PD or other synucleinopathies [117,118].

### 4.2. Mechanisms of GBA1-PD Development

Monoallelic (heterozygous) variants in the *GBA1* gene are linked to an increased risk of PD, suggesting an autosomal dominant (AD) pattern of inheritance. This risk arises through one of two possible mechanisms: either “haploinsufficiency”, where a single functional gene is not sufficient, or a “gain of function”, where the gene mutation causes the protein to acquire harmful properties [119,120].

#### 4.2.1. *GBA1* “Haploinsufficiency” as Mechanism for *GBA1*-PD Development

The term “haploinsufficiency” in the context of *GBA1*-PD indicates that the onset of PD is associated with a reduction in GCase activity and substrate accumulation. It is hypothesized that PD requires a substantially smaller reduction in GCase activity and a less significant build-up of substrate compared to GD, which is defined by a “loss-of-function” mechanism. However, if this were the primary mechanism, we would expect to see a higher number of patients with GD developing *GBA1*-PD.

Since the pioneering report linking GD and PD, a plethora of studies have been undertaken to comprehend the role of GCase and its substrates—specifically Glycosphingolipids (GSLs), such as glucosylceramide (GlcCer) and glucosylsphingosine (GlcSph)—in the pathogenesis of *GBA1*-PD. These investigations span a spectrum of subjects, including individuals with *GBA1*-PD and cases of iPD, and can be extended to various experimental models.

##### GCase Activity and Substrate Accumulation in *GBA1*-PD

Numerous studies have investigated whether a deficiency in GCase enzyme activity leads to substrate accumulation in patients with *GBA1*-PD as well as in cellular and animal models of the disease. In studies of human brains from patients with *GBA1*-PD, regions such as the SNpc, putamen, cerebellum, and amygdala showed significantly lower GCase activity compared to that of healthy controls, with the most significant reduction observed in the SNpc [121]. Autopsy studies of the brain revealed that GCase activity in the SN, putamen, and frontal cortex of patients with *GBA1*-PD, whether carrying a mild or severe *GBA1* variant, was reduced in comparison to that in iPD and healthy individuals [122]. This enzymatic reduction is also observable in dried blood spots from patients with *GBA1* mutations, where heterozygotes retain more activity than homozygotes and compound heterozygotes [123]. Midbrain dopaminergic neurons derived from induced pluripotent stem cells (iPSCs) of patients with PD carrying a heterozygous *GBA1* variant (p.L483P or p.N409S) exhibited decreased GCase activity alongside increased levels of GlcCer and alpha-synuclein [124]. Several studies have measured the substrate accumulation in dopaminergic cells established from iPSCs derived from heterozygous carriers of *GBA1* mutations. However, in another study, there were no significant differences between the levels of GlcCer accumulating in carriers of *GBA1* mutations versus control cells [125]. The same study documented the absence of GlcCer accumulation in Asp409Val/+ mice, with minor GlcSph accumulation. In several studies, the accumulation of GSLs has not been observed in the brains of patients with *GBA1*-PD [126,127], as well as accumulation of GlcCer has never been documented in the brains of patients with GD1 [126,128,129].

It is plausible that if the level of GCase activity, which regulates substrate amounts in lysosomes, plays a significant role in the development of PD, then abnormalities in genes that modulate GCase activity could also be associated with PD. This includes genes like *PSAP*, which encodes prosaposin, leading to the production of saposin C, a GCase activator, and SCARB2, which encodes LIMP-2, a receptor involved in the trafficking of GCase. However, findings regarding the association between SCARB2 polymorphisms and PD have been inconsistent. While two studies suggested a possible link [130,131]. Subsequent research did not confirm this association [132,133]. A recent study found no association between any saposin variants and PD [134], although another report highlighted that pathogenic missense variants in saposin C significantly increased the risk of developing PD. Clinically, patients carrying these variants exhibited typical motor symptoms of PD but did not show signs of cognitive impairment [135]. Additionally, Oji et al. [136] described three unrelated Japanese families with AD adult-onset PD, involving eight affected individuals, two of whom died, and their genetic material was not available for analysis. The remaining six presented with classic PD features from ages 33 to 60, including asymmetric onset, resting tremor, and an initially favorable response to dopaminergic therapy. Notably, two siblings from the second family, who harbored the PSAP mutation, displayed extrapyramidal signs, such as tremors and muscular rigidity, yet they were not definitively diagnosed with PD. In accordance with the role of PSAP in PD pathogenesis, a recent publication showed changes in PSAP levels in patients with PD [137]. Moreover, mice lacking PSAP in dopamine neurons develop lipid dyshomeostasis and PD-like phenotypes. Conversely, PSAP overexpression in rodents counteracts the experimental PD.

##### GCase Activity, Substrate Levels, and PD Pathology

Across various settings, both with and without *GBA1* mutations, numerous studies have established how GCase activity and substrate levels correlate with alpha-synuclein accumulation and PD symptom severity, including in animal models. Rocha et al. [138] found elevated levels of GlcSph and diminished GCase activity in the SN and hippocampus of patients with PD. In experiments involving iPSC-derived dopamine neurons with either biallelic or monoallelic *GBA1* mutations, the pathology of alpha-synuclein was observed to be consistent across both groups despite a significant reduction in GCase activity in individuals with biallelic *GBA1* mutations [124]. In a mouse model of synucleinopathy, treatment with a GlcCer-synthase inhibitor reduced alpha-synuclein pathology and restored normal behavior [139]. In A53T transgenic mice, overexpression of GCase in the central nervous system (CNS) caused a 15% decrease in the level of cytosolic soluble alpha-synuclein fraction [140].

However, other studies utilizing cell and animal models have not successfully demonstrated an association between GCase activity and alpha-synuclein accumulation [141,142,143]. Notably, primary neurons and transgenic mouse models subjected to Conduritol B Epoxide (CBE), an irreversible inhibitor of GCase treatment, did not exhibit increased levels of total alpha-synuclein or the development of alpha-synuclein pathology. Instead, there was an enhancement of pre-existing alpha-synuclein pathology, marked by an increase in pathogenic phosphorylated alpha-synuclein (p-S129-alpha-synuclein) [144]. This phenomenon is not restricted to neuron-specific responses.

Findings from in vitro studies revealed alpha-synuclein accumulation without a decrease in GCase activity in neuronal cell lines overexpressing *GBA1* mutations (p.N409S or p.L483P) [141]. A comprehensive single-center study found only a 5% reduction in GCase activity in dried blood spots from fresh whole blood of patients with iPD compared to healthy controls [123]. Yet, other investigations did not find these differences. [51,145,146,147,148]. A longitudinal study monitoring GCase activity changes in patients with iPD found no significant deviations from healthy controls [51]. Similarly, a recent analysis also showed no significant differences in GCase activity in dried blood spots of frozen whole blood samples between de novo patients with iPD and healthy controls [149]. Research on heterozygous *GBA1* knock-in (*Gba1*D409V/+) mice revealed the accumulation of alpha-synuclein aggregates in the brain, a phenomenon not observed in heterozygous *GBA1* knockout (Gba1+/−) mice. Interestingly, both mouse models exhibited similarly reduced levels of residual GCase activity [142]. This outcome implies that the presence of a single mutant (misfolded) form of GCase (D409V) might contribute to alpha-synuclein aggregate accumulation, independently of GCase enzymatic activity. Additionally, given the findings of clinical studies, individuals carrying monoallelic *GBA1* mutations have a similar risk of developing PD as individuals with biallelic *GBA1* mutations despite retaining greater GCase activity [26,123,150]. In a study of 170 patients with PD (102 *GBA1*-PD, 38 *LRRK2*-PD, and 30 iPD) and 221 non-manifesting carriers (NMC) (129 *GBA1*-NMC, 45 *LRRK2*-NMC, 15 *GBA1*-*LRRK2*-NMC, and 32 healthy controls), GCase activity was lower among *GBA1*-PD, *GBA1*-NMC, and *GBA1-LRRK2*-NMC compared to the other groups of participants, with no correlation to clinical phenotype [151]. These findings suggest that low GCase activity does not fully explain the clinical phenotype or the risk of PD. However, the penetrance indicates that additional factors, yet to be identified, contribute to the development of *GBA1*-PD.

##### Potential GCase and Its Substrates Related Mechanisms of Alpha-Synuclein Accumulation

Theoretically, the build-up of GlcCer and GlcSph could lead to impaired autophagy and mitophagy, mitochondrial dysfunction, and, ultimately, the accumulation and aggregation of alpha-synuclein. This process may result in the death of dopaminergic cells and the onset of PD. Numerous studies have explored the potential mechanisms by which decreased GCase activity and increased substrate accumulation may contribute to alpha-synuclein aggregation.

Evidence from the literature indicates that GlcCer accumulation triggered by treatment with CBE, a GCase inhibitor [152], leads to alpha-synuclein accumulation and aggregation [138]. In vitro studies have demonstrated that elevated GlcCer levels stabilize the soluble oligomeric forms of alpha-synuclein, fostering its aggregation [153,154]. Additionally, GlcCer has been shown to convert high-molecular-weight physiological conformers of alpha-synuclein into assembly state intermediates in cell-free in vitro models without breaking down into free monomers [155]. Further in vitro research on dopaminergic neurons derived from patients with *GBA1*-PD with the p.N370S variant revealed that intracellular GlcCer accumulation destabilizes normal alpha-synuclein tetramers/multimers, leading to the formation of soluble, toxic alpha-synuclein assemblies. This effect can be reversed by reducing GlcCer levels [156].

The findings from these extensive studies are ambiguous and contradictory, and they do not definitively confirm GCase “haploinsufficiency” as a mechanism for *GBA1*-PD development. Further evidence casts doubt on the “haploinsufficiency” theory as a direct link between decreased GCase activity, substrate accumulation, and subsequent alpha-synuclein build-up leading to *GBA1*-PD. *GBA1* mutations have been identified as a risk factor for PD but are not disease-causing mutations per se, as not every carrier develops the disorder [7,157]. Moreover, both patients with GD1, characterized by reduced GCase activity and high substrate levels in visceral organs but not in the brain, and asymptomatic carriers of a single *GBA1* mutation, who have some reduced GCase level and little or no substrate elevation, exhibit similar PD risks [28]. Some mild *GBA1* variants, such as the p.N370S, are associated with a higher risk of developing PD (OR for PD risk: 11.40) [52] compared to certain severe variants, like p.V394L (OR ranges from 4.85 to 6.70) [52,55].

Overall, the theory of *GBA1* “haploinsufficiency”—which posits that decreased GCase activity and the subsequent accumulation of GSLs such as GlcCer and GlcSph within the dopaminergic neurons of the CNS are critical in the development of *GBA1*-PD—lacks substantial support from current research. Therefore, the effectiveness of treatment strategies that rely solely on enzyme replacement or substrate reduction, in our opinion, should be called into question.

#### 4.2.2. *GBA1* “Gain-of-Function” Mechanism of *GBA1*-PD Development

As explained above, impairment of the GSL degradation pathways does not compromise the systems responsible for alpha-synuclein (protein) degradation. Building on this foundation, the hypothesis suggests a “gain-of-function” mechanism, where mutant forms of GCase are recognized as misfolded.

Misfolded GCase places undue stress on the protein degradation system, consequently impeding the breakdown of alpha-synuclein. When proteins within the ER do not fold correctly, they are retained in the ER for further attempts to refold them, causing a condition known as ER stress. Cytoplasmic alpha-synuclein aggregates are also degraded by phagolysosomes.

This stress activates a response mechanism called the Unfolded Protein Response (UPR). The UPR detects the presence of incorrectly folded proteins in the ER and sends signals to the cell’s nucleus. This triggers a specific set of actions aimed at restoring balance, including enhancing the ER’s ability to fold proteins correctly. As part of the UPR, the cell initiates another process known as ER-associated degradation (ERAD). ERAD specifically targets the misfolded proteins for destruction. It does so by ubiquitinating them, which signals their elimination by the proteasome, a complex designed to break down proteins and known as the UPS [158,159,160,161,162].

Research on heterozygous *GBA1* knock-in (*Gba1*D409V/+) mice revealed the accumulation of alpha-synuclein aggregates in the brain, a phenomenon not observed in heterozygous *GBA1* knockout (*Gba1*+/−) mice [142]. This outcome implies that the presence of a single mutant (misfolded) form of GCase (D409V) might have contributed to alpha-synuclein aggregate accumulation. In experiments involving human dopamine neurons and Drosophila flies with mutations in the *GBA1* gene, activation of UPR was observed. This stress response was accompanied by an increased aggregation of alpha-synuclein, suggesting a link between ER stress and the degradation of alpha-synuclein [124,163,164].

It has been clearly demonstrated that mutant GCase molecules are identified as misfolded within the ER [165,166,167], where they trigger ER stress and initiate the UPR [168,169]. The misfolded GCase is subsequently degraded through ERAD [9]. Inhibition of GCase activity by CBE or in knockout animals leads to an increase in GlcCer levels without causing protein misfolding [152,170]. It was clearly shown that this condition does not trigger the UPR and ERAD, highlighting that the presence of misfolded mutant GCase is necessary for activating these cellular mechanisms [171]. The severity of the ER stress has been shown to correlate with the severity of PD symptoms [165,172,173,174,175].

Several E3 ligases have been identified as interacting with and facilitating the ubiquitination and subsequent proteasomal degradation of mutant GCase in the ERAD. Notably, mutant GCase engages with the parkin protein, which is responsible for its Lysine 48-linked polyubiquitination and proteasomal degradation [176]. There is also observed competition between mutant GCase and two other substrates of parkin, namely ARTS and PARIS [177]. ARTS, a proapoptotic protein, when accumulated, leads to cell death [178]. Conversely, PARIS acts as a significant transcriptional repressor of NRF-1 gene expression [179]. Impaired degradation of PARIS by parkin results in enhanced repression of PGC-1α, a key regulator of mitochondrial biogenesis, via the induction of genes regulated by PARIS. The diminished transcription of these genes in cells expressing mutant GCase has been documented [177], adversely affecting mitochondrial biogenesis and function. This disruption can potentially contribute to increased neuronal death.

Numerous studies have utilized Drosophila as a model organism to illustrate that when mutant human *GBA1* cDNAs are expressed in the dopaminergic neurons of these flies, it leads to the ERAD of mutant GCase. This triggers the UPR and results in PD-like features: the loss of dopaminergic neurons, decreased climbing capacity, disruptions in sleep patterns, and a shortened lifespan [164,172,174,180].

ALP is the second main protein degradation pathway in the cells. ER stress can activate the ALP, which helps to remove misfolded or aggregated proteins from the ER and alleviate stress. In this process, a protein is sequestered into autophagosomes, which then fuse with lysosomes, where the protein is degraded by lysosomal enzymes [162]. This pathway is particularly important for the degradation of aggregated or oligomeric forms of alpha-synuclein [115]. Once alpha-synuclein is delivered to lysosomes via autophagy, including macro-autophagy and chaperone-mediated autophagy (CMA), it is degraded by a variety of lysosomal enzymes. These include the cathepsins. Specific enzymes directly degrading alpha-synuclein within lysosomes have not been uniquely identified because lysosomes contain a mix of enzymes that work together to degrade proteins [181]. In the case of CMA, proteins like alpha-synuclein that have a KFERQ-like motif are recognized by the chaperone HSC70 and directed to the lysosome, where they interact with LAMP2A for translocation into the lysosome for degradation [182]. An autophagy-lysosomal disturbance in *GBA1*-PD is observed in relation to ER stress in dopaminergic neurons differentiated from iPSCs of patients with *GBA1*-PD carrying a heterozygous p.N409S variant [163]. Numerous studies have demonstrated that mutations in the *GBA1* gene are linked to lysosomal abnormalities and disruptions in the processes of autophagy across a variety of animal and cellular models [164,183,184]. Studies using human iPSCs harboring *GBA1* mutations have reported ALP dysfunction in dopaminergic neurons from patients with PD and further highlighted its role in PD-associated proteinopathy [124,163,185]. Inhibition of macroautophagy decreased autophagosome-lysosome fusion, and lower cathepsin D levels and activity, were observed in dopaminergic neurons and were shown to contribute to alpha-synuclein accumulation [186,187,188]. Recent studies have uncovered a potential link between GCase, alpha-synuclein, and CMA. This connection is highlighted by the discovery that mutant GCase can mislocalize to the lysosome surface, a phenomenon observed in the brains of individuals with *GBA1*-PD. Intriguingly, half of the mutant GCase found in lysosomes was located on their surface, a misplacement that relies on a specific pentapeptide motif in GCase. This motif normally targets proteins for CMA degradation, but the aberrant positioning of mutant GCase hampers CMA, leading to the build-up of substrates like alpha-synuclein that should otherwise be degraded [189].

Mitochondria are essential for energy production through oxidative phosphorylation and are deeply involved in regulating calcium homeostasis, membrane potential, apoptosis, and stress response [190]. Their dysfunction is increasingly recognized as a critical factor in the pathogenesis of PD, with numerous studies linking mitochondrial anomalies to the development of iPD [190,191,192,193], as well as to *GBA1*-PD [194]. As previously noted, PARIS significantly represses the transcription of PGC-1α, which is a crucial regulator of mitochondrial creation through stimulating the expression of the NRF-1 gene [179]. Additionally, there is competition between mutant (misfolded) GCase and PARIS, as both are substrates for parkin [177]. When parkin fails to degrade PARIS effectively, there is increased repression of genes controlled by PARIS. The reduced transcription of these genes in cells with mutant GCase has been observed and negatively impacts mitochondrial biogenesis and function [177]. This interference can potentially lead to higher rates of neuronal death. An impaired mitochondrial function that hindered mitophagy was observed in hippocampal neurons from mice with a heterozygous L444P mutation. Additionally, there was a noted decrease in both general autophagy and mitochondrial priming in the brain tissue of these mice, as well as in their cultured hippocampal neurons. Similar results were found in postmortem studies of the anterior cingulate cortex in patients with *GBA1*-PD. This suggests that carriers of *GBA1* mutations experience deficits in mitophagy that contribute to the mitochondrial impairments linked to PD. The authors deduced that the distinct impacts of mutant GCase on mitochondrial priming and the initiation of autophagy suggest a harmful gain of function by the mutant protein [195].

A substantial body of research supports the “gain-of-function” hypothesis (Figure 1, which posits that regardless of its enzymatic activity, the misfolding of GCase can initiate ER stress and engage both the UPS and the ALP. The chronic overburdening of these protein quality control systems can impair the degradation of alpha-synuclein, thereby accelerating its aggregation. This insight directs future research towards the concept of “GCase misfolding” as a potential key factor in the development of *GBA1*-PD, emphasizing the need for new therapeutic strategies that target this mechanism.

## 5. *GBA1*-PD Treatment

### 5.1. Enhancing GCase Activity and Reducing Its Substrate Levels

Efforts to validate the hypothesis that enhancing GCase activity and reducing its substrate levels in neurons through treatment could potentially slow, halt, or reverse the progression of *GBA1*-PD draw upon the research outlined in the previous section. This evidence lays the groundwork for a proposed therapeutic strategy (Table 3). Among the existing methods/agents proposed to treat *GBA1*-PD that can lead to an increase in GCase activity or a reduction in its substrate, some are in the pre-clinical phase, others are currently undergoing clinical trials, and initial clinical trials for some have already been completed.

### 5.2. Small-Molecule Chaperones (SMCs)

Small-molecule chaperones (SMCs) specifically target either active or alternative sites on enzymes, promoting their transit to the intended organelle. Recognized as a promising avenue for *GBA1*-PD therapy, SMCs are capable of binding to GCase ensnared in the ER due to disrupted ER-Golgi trafficking, facilitating its transfer to the lysosome [207]. This mechanism offers relief to the UPS and the ALP, burdened by excessive ERAD, thereby augmenting their capacity to diminish alpha-synuclein concentrations in cells. An additional noteworthy advantage of SMCs is their ability to traverse the blood-brain barrier (BBB) [208]. A suite of SMCs, including the repurposed medications isofagomine, ambroxol, bromhexine, diltiazem, and fluphenazine, have been identified as potential remedies for GD [209,210]. These compounds have been demonstrated to engage with multiple molecular pathways associated with the development of *GBA1*-PD, mitigating ER stress and enhancing the cellular protein regulation system through the refolding and lysosomal transport of mutant GCase in fibroblasts and neurons possessing *GBA1* mutations [164,211,212,213,214]. Ambroxol and isofagomine have been demonstrated to alleviate ER stress and ameliorate symptoms in *GBA1*-mutant Drosophila models [164,174,180]. While precise dosing of SMC like isofagomine is essential to ensure they function effectively as chaperones—correctly refolding misfolded GCase for lysosomal delivery without inhibiting its activity [215]—clinical trials have highlighted challenges in their therapeutic application. Notably, a Phase 2 clinical trial of isofagomine in adults with GD1 did not yield promising results. Despite the treatment being well-tolerated and increasing GCase activity, only one out of 18 patients showed a meaningful reduction in disease symptoms over six months. This outcome led Amicus Therapeutics to conclude that there was insufficient evidence to continue the development of isofagomine as a treatment for GD, effectively halting further clinical development (https://gaucherdiseasenews.com/chaperone-therapy/isofagomine/ accessed on 24 June 2024) despite the theoretical option to restart with different dosing and/or administration regimens.

#### 5.2.1. Ambroxol

Ambroxol, a cough suppressant and mucolytic with an excellent safety profile has been safely used since the 1970s [208,216]. In 2009, ambroxol was identified in high-throughput screening of a regulatory-approved compound collection as an enhancer of stability and residual activity of several misfolded GCase variants [211]. Ambroxol has been shown to cross the BBB [216,217]. Ambroxol exhibits its inhibitory effects predominantly at the neutral pH of the ER while acting as an activator of GCase within the acidic environment of lysosomes [211]. Treatment with ambroxol has been shown to significantly elevate GCase activity in fibroblasts derived from patients with GD [208] as well as in patients with *GBA1*-PD [213]. In dopaminergic neurons obtained from patients with PD with the p.Asn409Ser variant, ambroxol therapy has been found to increase GCase activity and reduce alpha-synuclein accumulation by ameliorating ALP deficiency [212]. Additionally, in mouse models expressing the heterozygous p.Leu483Pro mutation or those overexpressing human alpha-synuclein, ambroxol treatment leads to enhanced GCase activity and decreased levels of alpha-synuclein [218]. High-dose oral ambroxol administration has also been observed to augment GCase activity in the brains of WT non-human primates [219]. This body of experimental evidence has paved the way for the initiation of clinical trials first targeting patients with GD.

An open-label pilot study involving five patients with nGD demonstrated that high-dose oral ambroxol, in conjunction with enzyme replacement therapy (ERT), achieves cerebrospinal fluid (CSF) ambroxol concentrations amounting to 10–20% of serum levels, reduces CSF GlcSph levels, and ameliorates neurological symptoms [216]. In patients with GD1, ambroxol was safely tolerated and exerted a positive effect on GCase (NCT03950050) [220]. Earlier, a pilot open-label study with 12 adult patients with GD1 naïve to any disease-specific treatment showed improvements in disease parameters after patients received ambroxol for 6 months [221]. The most significant effect of ambroxol in the treatment of GD was shown by Zhan et al. Patients, 18 years and older, with GD who could not afford ERT or substrate reduction therapy (SRT) due to the prohibitive cost received a high dose (13 ± 4.0 mg/kg/day) of oral ambroxol during the long-term (6 years) period. Significant improvements were observed in hematologic and visceral signs (increased levels of hemoglobin and platelet counts; a decrease in the volume of the liver and spleen), as well as in disease essential biomarkers (a reduction in plasma chitotriosidase activity and GlcSph level) [222].

Based on this evidence, in several countries where ambroxol is approved and available, it has been prescribed off-label with ERT or SRT in patients with nGD [216,223,224].

The progression to clinical studies investigating the biological impacts of ambroxol therapy in PD represents a natural evolution from foundational research in molecular and animal models, as well as studies on ambroxol application in GD.

The observational clinical study, “Investigator-initiated-research (IIR) REGISTRY for the collection of real-world data on the safety and efficacy of ambroxol for patients with GD or *GBA1* carriers with PD” (NCT04388969), recruits patients and analyzes the data on the safety and efficacy of ambroxol. The study recently reported initial data [225] of 41 patients (25 females) at a median (range) age of 17 (1.5–74) from 13 centers: 11 with GD1 (four diagnosed with PD), 27 with nGD, and three *GBA1* mutation carriers with PD. The median (range) treatment period and maximum dose of ambroxol were 19 months (1–76 months) and 435 mg/day (75–1485 mg/day), respectively. One patient with GD2 died. No other serious/severe adverse events (AEs) were reported. Clinical benefits were reported in 25 patients, including a stable or improved neurological status, increased physical activity, and reduced fatigue. The study continues to recruit patients and further analyze the data on the safety and efficacy of ambroxol.

In a single-center, open-label, non-controlled clinical trial, Mullin et al. [217] assessed the safety, tolerability, CNS penetration, and effectiveness of ambroxol therapy in patients with PD with and without *GBA1* mutations. Seventeen patients (15 men; mean age, 60 years; 8 with *GBA1* mutations) completed the study and were included in the analysis. Ambroxol was undetectable in the blood serum and CSF at baseline. On day 186, the CSF ambroxol level was 156 ng/mL. The CSF GCase activity decreased by 19%. The ambroxol therapy was well-tolerated, with no serious AEs. An increase of 13% in the CSF alpha-synuclein concentration and an increase of 35% in the CSF GCase protein levels were observed. Within the largely acellular CSF, the GCase protein is free, which is in contrast to its normal intracellular lysosomal location. The increase in CSF alpha-synuclein can be interpreted as an increase in the extracellular export of protein from the brain parenchyma. The mean scores on part 3 of the MDS-UPDRS decreased (i.e., improved) by 6.8 points. These changes were observed in patients with and without *GBA1* mutations. The study results suggest that ambroxol therapy is safe and well-tolerated; CSF penetration and target engagement of ambroxol were achieved, and CSF alpha-synuclein levels were increased, implying increased extracellular export of alpha-synuclein from the brain parenchyma, movement disorders were improved.

In a single-center, open-label, non-controlled clinical trial, 18 patients with PD, including 8 with a *GBA1* variant, were treated with 1.26 g/day ambroxol for six months (NCT02941822). Ambroxol treatment increased the ambroxol level (showing BBB penetration), decreased GCase activity, and increased alpha-synuclein levels in the CSF [217]. Ambroxol treatment may reduce GCase activity due to its inhibitory action on GCase activity in acellular human CSF with a neutral pH, while it will increase GCase activity in brain tissue at an acidic pH, as reported in animal studies [218,219]. Increased CSF alpha-synuclein levels after ambroxol treatment can be interpreted as an enhanced extracellular export of alpha-synuclein from the brain parenchyma. In addition, motor disability is improved after ambroxol treatment regardless of the presence of a *GBA1* variant. Ambroxol *GBA1*-PD ongoing clinical trials are shown in Table 4.

#### 5.2.2. Other SMCs

A novel quantitative high-throughput screening methodology utilizing spleen extracts from patients with GD has led to the identification of two compound series that activate GCase and demonstrate chaperone activity. These include a series of pyrazolopyrimidine carboxamide derivatives and salicylic acid derivatives [214,226].

Specifically, a pyrazolopyrimidine derivative, NCGC758, has been shown to increase GCase activity and reduce glycolipid storage in macrophages differentiated from monocytes or iPSCs of patients with GD [227]. The therapeutic potential of NCGC758 was further evaluated in iPSC-derived dopaminergic neurons from patients with PD variants, including *SNCA*, *GBA1*, and *PARK9*, and in iPD models. This evaluation demonstrated that NCGC758 specifically augments GCase activity within lysosomes and reduces GlcCer and alpha-synuclein levels across these neuronal models of PD, irrespective of the PD etiology or genetic variants involved [185].

LTI-291, also known as BIA-28-6156, another pyrazolopyrimidine, is identified as a small-molecule GCase allosteric activator. A Phase 1B clinical trial focused on *GBA1*-PD involving BIA-28-6156 was recently conducted, as registered in the Dutch Trial Registry (Netherlands Trial Register) under the study number NTR6960 [228]. This trial was a randomized, double-blind, placebo-controlled study that included 40 participants with *GBA1*-PD. Participants received daily doses of 10, 30, or 60 mg of BIA-28-6156 or a placebo for twenty-eight consecutive days, with 10 participants per treatment group. Measurements were taken of glycosphingolipid (specifically GlcCer and lactosylceramide) levels in peripheral blood mononuclear cells, plasma, and CSF. Additionally, participants underwent a battery of neurocognitive tasks, assessments using the MDS-UPDRS, and the Mini-Mental State Examination. BIA-28-6156 demonstrated good tolerability, with no deaths or treatment-related serious adverse events reported and no participants discontinued due to adverse effects. The maximum concentration (Cmax) and area under the curve (AUC0–6) of BIA-28-6156 exhibited dose-proportional increases, with the free CSF concentrations mirroring the free plasma fraction. Pharmacologically active concentrations in the plasma and CSF were achieved and deemed sufficient to at least double GCase activity and elevations in intracellular GluCer were observed.

The potential clinical benefits of BIA-28-6156 will undergo further evaluation in a larger, long-term Phase 2 trial in GBA-PD1: “A Randomized, Double-Blind, Placebo-Controlled Study to Evaluate the Efficacy, Safety, Tolerability, Pharmacodynamics, and Pharmacokinetics of BIA 28-6156 in Subjects With PD With a Pathogenic Variant in the *GBA1* Gene” (NCT05819359), with an estimated study completion date of 31 July 2026, and an anticipated enrollment of 237 participants.

Similarly, NCGC607, a derivative of salicylic acid, has been shown to restore GCase activity and decrease the levels of GlcCer and GlcSph in both macrophages and dopaminergic neurons derived from iPSCs of patients with GD1 and GD2. In addition, this compound significantly reverses alpha-synuclein accumulation in dopaminergic neurons derived from iPSCs of patients with GD2, as well as in patients with GD exhibiting a PD phenotype [229].

Quinazoline derivatives have emerged as potent non-iminosugar GCase inhibitors that also possess chaperone activity, as identified through high-throughput screening efforts. [230,231,232] These derivatives have been noted for their selective efficacy in stabilizing GCase. Subsequent structure-activity relationship studies revealed that N-methylation of quinazoline compounds transforms them from inhibitors to activators of GCase [233].

The effectiveness of S-181, an N-methylated quinazoline compound, was evaluated in iPSC-derived dopaminergic neurons from patients with iPD and those carrying PD-related genetic variants, such as *GBA1*, *LRRK2*, *DJ-1*, or *PARKIN*. Additionally, its efficacy was tested in PD mouse models with a heterozygous *GBA1* variant. Treatment with S-181 was found to enhance GCase activity and reduce the accumulation of GlcCer and alpha-synuclein in both in vitro and in vivo models of PD [125].

The dynamic interplay between diminished GCase activity, the accumulation of its substrates, and the ensuing neurodegenerative phenomena presents substantial challenges to traditional therapeutic strategies that primarily target enzyme replacement or substrate diminution. This complexity is further accentuated by the phenomenon of GCase misfolding within the ER, precipitating an overload of this critical protein quality control system. Such accumulation instigates ER stress, thereby implicating the UPS and the ALP in the pathogenesis. This scenario underscores the necessity of a comprehensive therapeutic strategy that goes beyond merely restoring GCase function. It should also address a wider spectrum of cellular dysfunctions, including the degradation of pathogenic proteins such as alpha-synuclein. In this evolving therapeutic panorama, SMCs such as ambroxol have emerged as frontrunners, heralding a new era in the management of *GBA1*-PD. Ambroxol, in particular, embodies the therapeutic promise of SMCs through its ability to traverse the BBB and its demonstrable efficacy in early clinical evaluations, showcasing significant potential in modulating the underlying pathophysiological processes of *GBA1*-PD.

## 6. Conclusions

*GBA1*-PD represents a distinct and complex subset of PD characterized by unique clinical, pathological, and molecular features. The insights gained from studying *GBA1*-PD have underscored the significant role of GCase in the pathogenesis of this condition. Mutations in the *GBA1* gene are the most significant genetic risk factors for PD, leading to distinct patterns of disease onset, progression, and symptomatology.

Current approved treatment options for PD, including *GBA1*-PD, primarily offer symptomatic relief without addressing the underlying pathophysiological or etiological mechanisms of the disease. However, understanding the relationship between *GBA1* mutations and PD has opened new avenues for research and therapeutic interventions, focusing on specific molecular mechanisms. This insight particularly emphasizes the need for targeted treatment strategies. While treatment approaches focusing on enhancing GCase activity and reducing substrate levels have encountered significant challenges, SMCs have shown promise. Further research is needed to identify interventions that can effectively modify the disease course.

*GBA1*-PD provides a unique window into the broader spectrum of PD, offering opportunities for advancing our understanding of the disease mechanisms and developing targeted therapeutic approaches. The distinctive nature of *GBA1*-PD reaffirms its status as a unique clinical and pathobiological PD phenotype, necessitating specific management and research approaches to improve outcomes in affected individuals.

## Figures and Tables

**Figure 1 ijms-25-07102-f001:**
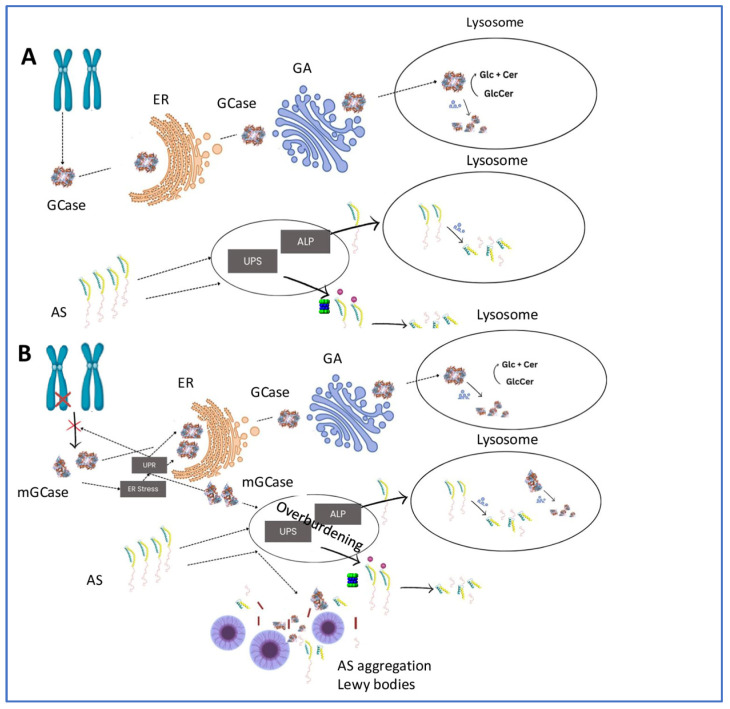
*GBA1* ‘Gain-of-function’ scenario for alpha-synuclein aggregation. (**A**) In healthy individuals, glucocerebrosidase (GCase) is synthesized in the endoplasmic reticulum (ER), properly folded, then trafficked to the Golgi apparatus (GA) for processing, and subsequently transported to the lysosome, where it degrades Glucosylceramide (GlcCer) to glucose (Glc) and ceramide (Cer). GCase is itself degraded in the lysosome by lysosomal enzymes. Cytoplasmic alpha-synuclein (AS) is degraded via two primary pathways: the ubiquitin–proteasome system (UPS) in the cytoplasm and the autophagy-lysosomal pathway (ALP) in the lysosome. (**B**) In *GBA1*-associated Parkinson’s disease (PD), mutant/misfolded GCase (mGCase) is recognized as misfolded in the ER, initiating ER stress and activation of the Unfolded Protein Response (UPR); mGCase is retained ER for further attempts to refold. When not successful, it undergoes retrotranslocation. Normal GCase (GCase) reaches the lysosome effectively and facilitates the degradation of GlcCer into Glc and Cer, and then itself is degraded in the lysosome by lysosomal enzymes. mGCase is directed to the UPS and ALP for degradation. Concurrently, the overburdening of the UPS and ALP due to the mGCase prevents AS from degrading at a normal pace, leading to its aggregation and the formation of Lewy bodies.

**Table 1 ijms-25-07102-t001:** Clinical Manifestations of GBA1-PD vs. iPD.

Clinical Manifestations	*GBA1*-PD	iPD	Ref.
Motor symptoms	Earlier onset; faster progression; often more severe	Gradual onset with slower progression	[7,8,19,20,21,22,23,24,25,26,27,28,29,30]
Non-motor symptoms	Higher prevalence; almost simultaneous occurrence of non-motor and motor symptoms	Typically, non-motor symptoms develop a longer period before the motor symptoms	[31]
Cognitive impairment	Greater impairment in working memory, executive function, visuospatial abilities, and greater risk of overt dementia	Common but generally less severe	[23,32,33,34,35,36,37]
Mood disorders	More frequent and severe mood disorders, including depression and anxiety	Common but generally less severe	[8,26,38,39,40,41]
Sleep disorders	More severe and more commonly reported, particularly REM RBD	Common but generally less severe	[34,35,36,37,38,39,42]
Olfactory dysfunction	More pronounced olfactory dysfunction	Common, but less pronounced	[38,39,40,43]
Autonomic symptoms *	Higher burden of autonomic symptoms, impacting quality of life	Common, with similar frequency but with a lower overall symptom burden	[38,40,41,43]
Response to standard treatment	Reduced effectiveness of standard PD treatments	Generally responds well to standard PD treatments	[7,8,19,20,21,22,23,24,25,26,27]

*GBA1*-PD—*GBA1*-associated Parkinson’s disease; iPD—idiopathic Parkinson’s disease; REM RBD—Rapid eye movement sleep behavior disorder; *—incl. constipation, bladder and erectile dysfunction, and orthostatic hypotension.

**Table 3 ijms-25-07102-t003:** Enhancing GCase activity and reducing its substrate levels as a treatment approach for *GBA1*-PD.

Approach	Description	Results/Comments	Ref.
Delivery of the rGCase through BBB	Linking rGCase to membrane-binding peptides: Tat, RDP, TTC, Tet1 (pre-clinical)	Potential RDP-GCase and Tat-GCase in crossing BBB	[196,197,198,199]
	Magnetic Resonance-guided Focused Ultrasound (MRgFUS) with microbubbles opens BBB for rGCase imiglucerase (clinical, NCT04370665)	Favorable safety profile; slight improvement in *GBA1*-PD motor and imaging signs	[200]
Preventing GCase degradation	HDACi blocks Hsp90 to GCase binding, preventing GCase ubiquitination and proteasomal degradation (pre-clinical)	Enhancement of GCase activity in GD fibroblast lines/could exacerbate PD due to the *GBA1* “gain-of-function” toxic mechanism	[201,202]
Gene therapy	Intracerebral injection of a *GBA1*-encoding AAV vector (pre-clinical)	Reducing alpha-synuclein accumulation; averting neuronal loss in SNpc	[140,203]
	Intraventricular injection of PR001, an AAV9 vector harboring the *GBA1* gene (pre-clinical)	reducing glycolipid substrates; ameliorating behavioral deficits	[204]
	Evaluating the effectiveness of intra-cisternal delivery of PR001 in patients with *GBA1*- PD (phase I/IIa non-randomized clinical trial, NCT04127578)	Ongoing: participants (estimated)—20; completion date (estimated)—June 2029	Ongoing
Substrate reduction therapy	Evaluating the effectiveness of venglustat in treating *GBA1*-PD (phase II clinical trial, NCT02906020)	Favorable safety profile; failed to exhibit any advantageous therapeutic effect compared to placebo	[205,206]

rGCase—recombinant beta-glucocerebrosidase; BBB—blood-brain barrier; Tat—Tat-peptide; RDP—rabies glycoprotein-derived peptide; TTC—tetanus toxin’s binding domain; Tet1—tetanus-like peptide; HDACi—Histone deacetylase inhibitors; GD—Gaucher disease; SNpc—Substantia Nigra pars compacta.

**Table 4 ijms-25-07102-t004:** Ambroxol *GBA1*-PD ongoing clinical trials.

No.	Study Name	Registration	Participants Enrolment (Estimated)	Completion Date (Estimated)
1	An Open-Label Pilot Study for Assessing the Safety and Efficacy of High-Dose Ambroxol in Newly Diagnosed *GBA1* PD	NCT06193421	40	April 2025
2 *	Ambroxol as a Novel Disease Modifying Treatment for PD Dementia. A double-blind, randomized, placebo-controlled trial to test whether ambroxol treatment can reduce the rate of cognitive decline in PD patients with dementia	NCT02914366	55	December 2025
3	GRoningen Early-PD Ambroxol Treatment (GREAT) Trial: A Randomized, Double-blind, Placebo-controlled, Single-center Trial With Ambroxol in PD Patients With a *GBA* Mutation	NCT05830396	80	July 2025
4	Ambroxol as a Disease-modifying Treatment to Reduce the Risk of Cognitive Impairment in *GBA*-associated PD. A Multicenter, Randomized, Double-blind, Placebo-controlled, Phase 2 Trial	NCT05287503	65	December 2024
5	IIR REGISTRY for the Collection of Real-World Data on the Safety and Efficacy of Ambroxol for Patients With GD or *GBA* Carriers With PD	NCT04388969	300	January 2024
6 *	Ambroxol to Slow Progression in Parkinson’s Disease	NCT05778617	330	March 2027

PD—Parkinson’s disease; GD—Gaucher disease. * The study also includes subjects with wild-type. *GBA1*.

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
