# Peer review of "GBA1-Associated Parkinson’s Disease Is a Distinct Entity"

_ijms, 2024, doi:10.3390/ijms25137102_

Round 1

Reviewer 1 Report

Comments and Suggestions for Authors

1. Simplify the description of iPD: Since researchers studying PD are already familiar with iPD and it is not the focus of this article, it is recommended to briefly describe iPD in the introduction and not mention it again in the sections on clinical manifestations and epidemiology.

2. Present the comparison between GBA1-PD and iPD in a table format: This will provide a clear and concise overview, avoiding repetition. For example, the description in lines 119-120 has already been mentioned in lines 80-84.

3. Correct the extra dot in "2.2. . Clinical manifestations: GBA1-PD vs iPD".

4. Simplify lines 251-274: As the focus of this article is GBA1-PD, it is unnecessary to delve into the epidemiology of iPD.

5. Provide literature references for the two potential mechanisms of GBA1-PD: "Haploinsufficiency" and "gain of function".

Author Response

We sincerely thank the reviewer for the insightful comments and valuable suggestions. The manuscript has been thoroughly revised and improved accordingly.

  1. Comment: Simplify the description of iPD: Since researchers studying PD are already familiar with iPD and it is not the focus of this article, it is recommended to briefly describe iPD in the introduction and not mention it again in the sections on clinical manifestations and epidemiology.

Response: We agree with your suggestion. Accordingly, we have added a brief description of idiopathic Parkinson Disease (iPD) in the introduction (line 58) and have omitted mentions of "iPD" wherever possible in the sections on clinical manifestations and epidemiology.

  1. Comment: Present the comparison between GBA1-PD and iPD in a table format: This will provide a clear and concise overview, avoiding repetition. For example, the description in lines 119-120 has already been mentioned in lines 80-84.

Response: We have replaced the text with a clear and concise table that provides a comprehensive comparison between GBA1-PD and iPD, ensuring better readability and avoiding redundancy. (Table 1. Lines 101-104)

  1. Comment: Correct the extra dot in "2.2. . Clinical manifestations: GBA1-PD vs iPD".

Response: We corrected this. (Line 80)

  1. Comment: Simplify lines 251-274: As the focus of this article is GBA1-PD, it is unnecessary to delve into the epidemiology of iPD.

Response: We have shortened the long description into one clear paragraph to maintain the focus on GBA1-PD and avoid delving into the epidemiology of iPD. (Lines 220-226).

  1. Comment: Provide literature references for the two potential mechanisms of GBA1-PD: "Haploinsufficiency" and "gain of function".

Response: We have added literature references for the two potential mechanisms of GBA1-PD: "haploinsufficiency" and "gain of function": references 119, 120. (Line 310)

  1. Veita, RA, Caburet, S. and Birchler, JA. Mechanisms of Mendelian dominance”. Clinical Genetics. 2018;93:419–428. DOI: 10.1111/cge.13107
  2. Carvill, GL, Matheny, T, Hesselberth, J. and Demarest, S. “Haploinsufficiency, Dominant Negative, and Gain‐of‐Function Mechanisms in Epilepsy: Matching Therapeutic Approach to the Pathophysiology” Neurotherapeutics (2021) 18:1500–1514. https://doi.org/10.1007/s13311-021-01137-z

Reviewer 2 Report

Comments and Suggestions for Authors

I was happy to read this informative and timely review.  In general it was very up to date and valuable to someone interested in PD and not an expert on GBA.  I hope the authors will find these comments constructive. 

1.  In the abstract, the authors state that the "gain of function" model is the most plausible.  While it is clear that the UPR is highly relevant to certain GBA mutations, the rate of GBA synthesis may not be that high as it is a long lived enzyme.  I agree with the authors that we cannot forget the UPR and its consequences for Synuclein clearance.  However GBA activity is also needed for normal autophagic degradation of synuclein.  Thus I would tone down the abstract on this specific point.

2.  Line 99.  Alpha synuclein aggregates would be expected to overlap with lysosomal GCase.  I would omit the word ACTUALLY and instead add, presumably in lysosomes.  Also, the reader is interested in learning about penetrance of GBA mutations in relation to PD at this point in the article.

3.  FIg. 1.  It would be much more informative for the reader to see the locations of the most common variants.  "Risk PD variant" was unclear?  Perhaps the 11 exons are less relevant than residue number in the polypeptide chain?  Since the authors stress a Gain of function paradigm, it might be useful to indicate (in relation to phenotype) which mutations have the most problems folding as opposed to those that simply lack GCase activity

4.  Line 398 please add the word Trafficking to describe LIMP2 as a receptor

5.  Line 405 what was the mutation in this family

6.  Overexpression of GBA (line 438), even a mutant, is not the same as that protein at endogenous levels, as it will still increase activity to some extent thus one might not expect a decrease in GBA activity

7.  Line 459. "...low GCase activity does not explain the clinical phenotype or risk for PD". This is true but the penetrance tells us that additional yet to be identified factors contribute to GBA PD.  Please state this.

8.  :Line 495.  "The effectiveness of treatment strategies that rely solely on enzyme replacement or substrate reduction is called into question."  This is perhaps too strong.  Until we understand the penetrance paradox, such a statement is premature.  Again, matching GBA mutations that cause the greatest UPR challenge with PD probability might be most useful herein.

9.  Line 502 please add cytoplasmic synuclein as aggregates will also be degraded by phagolysosomes.

10.  Line 557.  Evidence for CMA in synuclein degradation is weak and not yet compelling.

11.  Line 597.  Loss of mitophagy and initiation of autophagy need not be due to a gain of function process.  It could be that GCase deficient lysosomes can't do those.

12.  Figure 2 is confusing.  GCase is trying to fold in the lumen of the ER.  When it cannot, it is exported to the cytosplasm by ERAD and degraded.  Synuclein aggregates are degraded in lysosomes; monomers in the cytoplasm.  The localizations are not clear here. 

13.  Line 629.  Refs 215 and 216 compare completely different treatment regimens.  One is enzyme replacement with one dose and the other, a small molecule that hopefully decreases substrates.  They cannot be compared.  Better to describe what has been tried and take the reader through the table. 

14.  Line 722 Might ambroxal be indicing lysosome exocytosis?  Also lines 722-725 seem repeated between 738 and 740

15.  Table 2.  No. 2.  Please state if these were GBA carriers

16.  Line 796.  Did the quinazoline have effects on non-GBA mutant carriers?

17.  Please speculate why Ambroxal was more effective than isofagomine.  Also, did success correlate with particular mutations--i.e. those easier to refold and no success with simple activity deficient forms?

18.  Thank you for this compendium which was surely a great deal of work

Author Response

We thank the reviewer for the positive feedback and thoughtful comments. Below, we address each of the comments.

  1. Comment: In the abstract, the authors state that the "gain of function" model is the most plausible.  While it is clear that the UPR is highly relevant to certain GBA mutations, the rate of GBA synthesis may not be that high as it is a long lived enzyme.  I agree with the authors that we cannot forget the UPR and its consequences for Synuclein clearance.  However GBA activity is also needed for normal autophagic degradation of synuclein.  Thus I would tone down the abstract on this specific point.

Response: In the abstract, we stated: “Continued research is advancing our understanding of how these mechanisms contribute to the development and progression of GBA1-PD, with the 'gain of function' mechanism appearing to be the most plausible.” This statement reflects the understanding of the authors based on current literature. We believe this adequately represents our findings and the prevailing perspective, and thus we do not feel it should be toned down.

  1. Comment: Line 99.  Alpha synuclein aggregates would be expected to overlap with lysosomal GCase.  I would omit the word ACTUALLY and instead add, presumably in lysosomes.

Response: The comment has been addressed: “… there is a co-localization of the mutant GCase along with alpha-synuclein, presumably in lysosomes, … (lines 85-86).

Comment: Also, the reader is interested in learning about penetrance of GBA mutations in relation to PD at this point in the article.

Response: We acknowledge the relevance of penetrance to the clinical aspects of GBA1-PD. We have addressed the topic of penetrance in the manuscript (lines 260-270), recognizing its significance in population studies. Additionally, based on the recommendation of reviewer 1, we have condensed the clinical section and now present it in a table format. Moving the discussion of penetrance to the clinical section at this stage would be challenging. We guess that the current organization of the manuscript maintains clarity and coherence.

  1. Comment: FIg. 1. It would be much more informative for the reader to see the locations of the most common variants.  "Risk PD variant" was unclear? 

Response: Since it is extremely challenging to reflect even the most essential details of all GBA1 variants (incl. location) within a single figure, we have decided to remove the figure from the manuscript. Instead, we redirect the readers to open sources such as “ClinVar” and the “GBA1 Variant Browser”. These sources provide easily accessible, detailed visual information on all GBA1 variants and allow users to navigate and find specific variant characteristics. In parallel, we added a table (Table 2) with detailed information about most commonly associated with Parkinson disease (PD) GBA1 variants, including Risk PD variants. (Table 2. Lines 177-183)

Comment: Perhaps the 11 exons are less relevant than residue number in the polypeptide chain?  Since the authors stress a Gain of function paradigm, it might be useful to indicate (in relation to phenotype) which mutations have the most problems folding as opposed to those that simply lack GCase activity (Please advise what corrections to the figure should we do).

Response: We have added a paragraph providing a brief overview of all known GBA1 variants, including their pathogenicity and types of mutations. (lines 114-128) This addition will enhance clarity for readers. In this section, we highlight that unlike Gaucher disease, where there is a direct and significant correlation between the decrease in enzyme activity and disease severity, such a correlation is not evident in Parkinson disease (PD). This suggests that other characteristics of GCase, such as proper folding or misfolding, might play a crucial role in PD development. Most GBA1 variants are missense mutations, which involve a single nucleotide substitution, but their PD phenotypic spectrum ranges from no-risk to high-risk of severe PD. While measuring enzyme activity is straightforward, assessing the degree of enzyme misfolding is more complex. This represents a significant gap in GBA1-associated PD research, and we hope future studies will address this important area.

  1. Comment: Line 398 please add the word Trafficking to describe LIMP2 as a receptor

Response: The word “trafficking” was added: This includes genes like PSAP, which encodes prosaposin leading to the production of saposin C, a GCase activator, and SCARB2, which encodes LIMP-2, a receptor involved in the trafficking of GCase. (Lines 349-352)

  1. Comment: Line 405 what was the mutation in this family

Response: Oji et al. described three unrelated Japanese families with autosomal dominant adult-onset Parkinson disease. In these families, two variants in the intronic regions of the PSAP saposin D domain were identified: NM_002778.4(PSAP):c.1351-14A>G (rs4747203) and NM_002778.4(PSAP):c.1432-22C>T (rs885828).

  1. Comment: Overexpression of GBA (line 438), even a mutant, is not the same as that protein at endogenous levels, as it will still increase activity to some extent thus one might not expect a decrease in GBA activity

Response: We appreciate your comment and the nuanced perspective on the overexpression of GBA1.

  1. Comment: Line 459. "...low GCase activity does not explain the clinical phenotype or risk for PD". This is true but the penetrance tells us that additional yet to be identified factors contribute to GBA PD.  Please state this.

Response: The comment has been addressed. Here is a corrected text: “These findings suggest that low GCase activity does not fully explain the clinical phenotype or risk for PD. However, the penetrance indicates that additional, yet to be identified, factors contribute to the development of GBA-associated PD.” (Line 411-414)

  1. Comment: Line 495.  "The effectiveness of treatment strategies that rely solely on enzyme replacement or substrate reduction is called into question."  This is perhaps too strong.  Until we understand the penetrance paradox, such a statement is premature.  Again, matching GBA mutations that cause the greatest UPR challenge with PD probability might be most useful herein.

Response: Some investigators still believe in haploinsufficiency (loss of function) as the key underlying mechanism leading to GBA1-PD, in contrast, we are convinced that the pathological mechanism is gain of function due to the misfolding of the protein, hence, “called into question” is not that strong. Nevertheless, as we are aware that science can be multi-faceted, we have modified the sentence and replaced the “is” by “in our opinion should be”. (Lines 448-450)

  1. Comment: Line 502 please add cytoplasmic synuclein as aggregates will also be degraded by phagolysosomes.

Response: The sentence was added:  Cytoplasmic alpha-synuclein aggregates are also degraded by phagolysosomes. (Lines 459-460)

  1. Comment: Line 557.  Evidence for CMA in synuclein degradation is weak and not yet compelling.

Response: Thank you for your insightful comment. In our review, we aim to present and discuss the current literature related to CMA in alpha-synuclein degradation. Our intention is to provide an overview of existing studies without making definitive judgments on the scientific validity of this pathway. We acknowledge that the evidence for CMA in synuclein degradation is still evolving.

  1. Comment Line 597.  Loss of mitophagy and initiation of autophagy need not be due to a gain of function process.  It could be that GCase deficient lysosomes can't do those.

Response: We understand that the loss of mitophagy and the initiation of autophagy might not solely result from a gain of function process. It is important to note that in (heterozygous) carriers, there are no completely GCase-deficient lysosomes; rather, there are lysosomes with decreased GCase activity.

  1. Comment: Figure 2 is confusing.  GCase is trying to fold in the lumen of the ER.  When it cannot, it is exported to the cytosplasm by ERAD and degraded.  Synuclein aggregates are degraded in lysosomes; monomers in the cytoplasm.  The localizations are not clear here.

Response: Based on the comments, we have revised the figure (entitled Figure 1 in the revised version) to more accurately represent the folding of GCase in the ER lumen, its export to proteasome degradation when misfolded, and the degradation of alpha-synuclein aggregates in lysosomes and monomers in the cytoplasm. The updated figure should now provide clearer localization and pathways. (in revised version - Figure 1, lines 559-574)

  1. Comment: Line 629.  Refs 215 and 216 compare completely different treatment regimens.  One is enzyme replacement with one dose and the other, a small molecule that hopefully decreases substrates.  They cannot be compared.  Better to describe what has been tried and take the reader through the table.

Response: Many thanks for pointing this out. We have removed the sentence from the text (see line 584);

  1. Comment: Line 722 Might ambroxal be indicing lysosome exocytosis? Comment: Also lines 722-725 seem repeated between 738 and 740.

Response: We have removed the redundancy: “Ambroxol is a pH-dependent inhibitory SMC that binds to the active site of the GCase. Binding enables transportation to the lysosome and elution of free active enzymes under acidic conditions. Therefore, in acellular CSF, ambroxol will bind to and inhibit free GCase. However, in tissues, including those in the brain, ambroxol will increase intracellular GCase activity.” Now, the revised manuscript includes only this sentence: “Ambroxol treatment may reduce GCase activity due to its inhibitory action in acellular human CSF with a neutral pH, while it will increase GCase activity in brain tissue with an acidic pH, as reported in animal studies.” (Lines 689-691) Regarding the potential of Ambroxol to induce lysosome exocytosis, we do not have information directly indicating its role in promoting lysosome exocytosis.

  1. Comment: Table 2.  No. 2.  Please state if these were GBA carriers

Response: Thank you for the important comment. All the studies listed in the table (entitled Table 4 in the revised version) are ambroxol studies in relation to PD and LBD, in two of the studies marked by * not all the subjects were GBA1 carriers and we have clarified it in the footnote (see lines 696-698).

  1. CommentLine 796.  Did the quinazoline have effects on non-GBA mutant carriers?

Response: The potential efficacy of quinazoline for non-GBA1-PD has indeed been studied (e.g., Ahmad et all, 2024, below). However, our review focuses specifically on GBA1-PD, and we would prefer not to delve into this matter.

Ahmad I, Khalid H, Perveen A, Shehroz M, Nishan U, Rahman FU, Sheheryar, Moura AA, Ullah R, Ali EA, Shah M, Ojha SC. Identification of Novel Quinolone and Quinazoline Alkaloids as Phosphodiesterase 10A Inhibitors for Parkinson's Disease through a Computational Approach. ACS Omega. 2024 Mar 26;9(14):16262-16278. doi: 10.1021/acsomega.3c10351. PMID: 38617664; PMCID: PMC11007772.

  1. Comment: Please speculate why Ambroxal was more effective than isofagomine.  Also, did success correlate with particular mutations--i.e. those easier to refold and no success with simple activity deficient forms?

Response: The experiments involving ambroxol and isofagomine were conducted by other investigators. We document their results but are not in a position to speculate on the differences in clinical efficacy between the two chaperone molecules, which seem to be identical in the in-vitro studies. While the development of isofagomine stopped due to disappointing results of the phase 2 clinical trials, ambroxol has been studied in several IIRs simply because of its availability as an over the counter drug, allowing broader experiments. Because isofagomine is no longer in development we do not expect future studies, nor do we think there is any relationship to specific mutations.

  1. Thank you for this compendium which was surely a great deal of work

We thank the reviewer for the kind words and appreciation. We do hope that the present review contributes meaningfully to the understanding of GBA1-associated Parkinson Disease.